Dongsha Atoll is an important stepping-stone that promotes regional genetic connectivity in the South China Sea

Liu Shang Yin Vanson 1 2 syvliu@mail.nsysu.edu.tw
Green Jacob 3 4
Briggs Dana 3
Hastings Ruth 3
http://orcid.org/0000-0002-4699-6056 Jondelius Ylva 3
Kensinger Skylar 3 5
Leever Hannah 3
Santos Sophia 3
http://orcid.org/0000-0002-2370-9122 Throne Trevor 3
Cheng Chi 2
http://orcid.org/0000-0003-4260-5625 Madduppa Hawis 6
http://orcid.org/0000-0001-6339-4340 Toonen Robert J. 7
http://orcid.org/0000-0002-0371-5621 Gaither Michelle R. 8
http://orcid.org/0000-0001-8580-3651 Crandall Eric D. 3 9 eric.d.crandall@gmail.com
1 Dongsha Atoll Research Station, College of Marine Sciences, National Sun Yat-sen University , Kaohsiung City , Taiwan
2 Department of Marine Biotechnology and Resources, National Sun Yat-Sen University , Kaohsiung City , Taiwan
3 School of Natural Sciences, California State University , Monterey Bay, California , United States
4 Department of Biological and Environmental Science, University of Rhode Island , Kingston, Rhode Island , United States
5 Department of Molecular, Cellular and Developmental Biology, University of California, Santa Cruz , Santa Cruz, California , United States
6 Department of Marine Science and Technology, Institut Pertanian Bogor , Bogor , Indonesia
7 Hawai‘i Institute of Marine Biology, University of Hawai‘i at Mānoa , Kane‘ohe, Hawai‘i , United States
8 Department of Biology, University of Central Florida , Orlando, Florida , United States
9 Department of Biology, Pennsylvania State University , University Park, Pennsylvania , United States
Jeffery Nicholas
Electronic publication date: 2021 Aug 31
Publication date: 2021
Volume: 9
Electronic Location ID: e12063
Received 2021 Apr 19; Accepted 2021 Aug 4
Copyright: © 2021 Liu et al.
Copyright year: 2021
Copyright holder: Liu et al.
License: This is an open access article distributed under the terms of the Creative Commons Attribution License, which permits unrestricted use, distribution, reproduction and adaptation in any medium and for any purpose provided that it is properly attributed. For attribution, the original author(s), title, publication source (PeerJ) and either DOI or URL of the article must be cited.
License URL: https://creativecommons.org/licenses/by/4.0/

Keywords: Indo-Pacific, Migration models, Larval dispersal, Marine connectivity, Phylogeography, Marine metapopulations, Stepping-stones

Funding: Ministry of Science and Technology, Taiwan (MOST) and National Science Foundation (NSF) 109-2119-M-110 -003 and NSF-DEB#1457848 This research was financially supported by the Ministry of Science and Technology, Taiwan (MOST) and the National Science Foundation (NSF) under 109-2119-M-110 -003 and NSF-DEB#1457848, respectively. The funders had no role in study design, data collection and analysis, decision to publish, or preparation of the manuscript.

==============================
Background

Understanding region-wide patterns of larval connectivity and gene flow is crucial for managing and conserving marine biodiversity. Dongsha Atoll National Park (DANP), located in the northern South China Sea (SCS), was established in 2007 to study and conserve this diverse and remote coral atoll. However, the role of Dongsha Atoll in connectivity throughout the SCS is seldom studied. In this study, we aim to evaluate the role of DANP in conserving regional marine biodiversity.

Methods

In total, 206 samples across nine marine species were collected and sequenced from Dongsha Atoll, and these data were combined with available sequence data from each of these nine species archived in the Genomic Observatories Metadatabase (GEOME). Together, these data provide the most extensive population genetic analysis of a single marine protected area. We evaluate metapopulation structure for each species by using a coalescent sampler, selecting among panmixia, stepping-stone, and island models of connectivity in a likelihood-based framework. We then completed a heuristic graph theoretical analysis based on maximum dispersal distance to get a sense of Dongsha’s centrality within the SCS.

Results

Our dataset yielded 111 unique haplotypes across all taxa at DANP, 58% of which were not sampled elsewhere. Analysis of metapopulation structure showed that five out of nine species have strong regional connectivity across the SCS such that their gene pools are effectively panmictic (mean pelagic larval duration (PLD) = 78 days, sd = 60 days); while four species have stepping-stone metapopulation structure, indicating that larvae are exchanged primarily between nearby populations (mean PLD = 37 days, sd = 15 days). For all but one species, Dongsha was ranked within the top 15 out of 115 large reefs in the South China Sea for betweenness centrality. Thus, for most species, Dongsha Atoll provides an essential link for maintaining stepping-stone gene flow across the SCS.

Conclusions

This multispecies study provides the most comprehensive examination of the role of Dongsha Atoll in marine connectivity in the South China Sea to date. Combining new and existing population genetic data for nine coral reef species in the region with a graph theoretical analysis, this study provides evidence that Dongsha Atoll is an important hub for sustaining connectivity for the majority of coral-reef species in the region.

Introduction

With coral reefs and their communities in accelerating global decline (Hughes et al., 2003, 2018), governments around the world have established Marine Protected Areas (MPAs) as a way to counteract this trend (Selig & Bruno, 2010). It is now quite clear that MPAs, especially when planned as part of a network, can increase biodiversity, enhance the biomass of fished species, and promote ecosystem resilience (Emslie et al., 2015; Mellin et al., 2016; Sala et al., 2021). When properly sized and spaced, a network of MPAs can protect coral reef populations, and an adequate fraction of their larval offspring that disperse to nearby MPAs, ensuring both long-term community persistence and fisheries spillover (Gaines et al., 2010; Krueck et al., 2017). However, while there are a few robust examples of MPA networks planned at the regional level (Gleason et al., 2013; Emslie et al., 2015), most MPAs are singletons implemented by regional governments for local benefit only (Gaines et al., 2010).

Singleton MPAs are still valuable for the conservation and resilience of local coral reef communities and maintaining local fisheries through larval export and adult spillover (Lester et al., 2009; Mellin et al., 2016). However, it remains crucial to evaluate, post hoc, their role in conserving regional marine biodiversity. Does the MPA serve as a useful intergenerational stepping-stone that can augment demographic and genetic connectivity among regional reefs (McCook et al., 2009)? Coalescent samplers are a family of population genetic models that allow a relatively quick and inexpensive way to make such an evaluation (Crandall, Treml & Barber, 2012; Selkoe et al., 2016; Crandall et al., 2019b). In a likelihood-based model selection framework, such methods can distinguish between models of effective panmixia (high regional gene flow) and metapopulation models in which larvae disperse only to nearby populations (stepping-stone model) or to all sampled populations (island model; Beerli & Palczewski, 2010).

Understanding the role of a given MPA in region-wide connectivity patterns requires genetic data from many species from inside the MPA and from the surrounding populations. Fortunately, a working group of the Diversity of the Indo-Pacific Network has compiled a genetic database comprising over 38,000 individuals from more than 230 species (DIPnet; Crandall et al., 2019a), stored within the Genomic Observatories Metadatabase (GEOME; Deck et al., 2017; Riginos et al., 2020). This open database provides the opportunity to test various biogeography, speciation, and connectivity hypotheses across a wide taxonomic swath of species comprising Indo-Pacific communities, without the need to sample every location and every species by a single research group.

Located 340 km southeast of Hong Kong and 850 km southwest of Taipei, Dongsha Atoll, with an area of about 600 km2, is the largest and oldest atoll in the South China Sea (SCS; Fig. 1), (Dai, 2004). The atoll is of strategic political importance in that it sits in the Taiwan Strait, a major trade route between East and Southeast Asia. The atoll also hosts important coral reef habitats in the northern South China Sea, providing necessary fishing opportunities for the people of China, Taiwan, and Vietnam (Dai, 2004). For this reason, the Dongsha Atoll National Park (DANP) was established in 2004 by the Taiwanese government. The DANP encompasses 3,537 km2 of marine habitat (Cheng et al., 2020), and its establishment has helped to mitigate the impact of massive bleaching events in 1998 and overfishing at the atoll, which were negatively impacting coral cover and biodiversity (Fang, 1998; Soong, Dai & Lee, 2002; Dai, 2004). With proper enforcement and regional cooperation in place since 2007, a general survey of marine resources in the DANP in 2011 showed that both terrestrial and marine ecological resources are gradually recovering (Dai, 2012). Contradicting these results, Cheng et al. (2020) detected a dramatic 34% decrease in coral coverage since the DANP was established, with an alarming reduction in the abundance of branching corals indicating an overall simplification of habitat types. Although the effect of the establishment of the DANP on community biodiversity and health is still under debate, how these conservation efforts might have impacted other reefs in the South China Sea and beyond is poorly understood.

Figure 1 Sampling map.

(A) Map of all sampled locations. Outer rings are colored by which species were sampled at a locality, while inner circles are keyed to regional colors in Fig. 3. (B) Inset of South China Sea showing coral reefs in red following Zhao et al. (2016) and Dorman et al. (2016). Summer surface circulation patterns (solid line with arrows; winter circulation is roughly reversed) and Kuroshio Current intrusion (dotted line with arrows) follow Hu et al. (2000). Open circles are colored by species, and give estimates of maximum larval dispersal distances given mean summer surface current speed of 18.7 km/day (Hu et al., 2000) and PLD s given in Table 1. Solid circles indicate species for which a stepping-stone model was selected, and dashed lines indicate species for which panmixia was selected. Maps were generated from the public domain Natural Earth raster with the Cartopy v0.11.2 package for Python (Met Office, 2014).

Table 1 Species names, estimated pelagic larval durations, and primers and annealing temperatures used for PCR.

Scientific name (dataset citation)	Common name	Maximum pelagic larval duration (days)	mtDNA Locus (base pairs) & Primers used	Annealing temp (°C)	
Acanthurus japonicas (DiBattista et al., 2016)	Japanese surgeonfish	62 (for A. triostegus; McCormick, 1999)	Cytochrome-B (491)	62	
			Cytb9/Cytb7 (DiBattista et al., 2016)		
Centropyge vrolikii (DiBattista et al., 2012)	Pearlscale angelfish	29 (Thresher & Brothers, 1985)	Cytochrome-B (575)	58	
			CLFM_FOR/CLFM_REV (DiBattista et al., 2012)		
Chaetodon auriga (DiBattista et al., 2015)	Threadfin butterflyfish	48 (Wilson & McCormick, 1999)	Cytochrome-B (668)	56	
			Cytb9/Cytb7 (DiBattista et al., 2013)		
Chaetodon lunulatus (Waldrop et al., 2016)	Oval butterflyfish	35 (Soeparno et al., 2012)	Cytochrome-B (605)	50	
			Cytb9/Cytb7 (Waldrop et al., 2016)		
Ctenochaetus striatus	Striated surgeonfish	59 (Wilson & McCormick, 1999)	Control Region (316)	50	
			CR-A/CR-E (Lee et al., 1995)		
Dascyllus aruanus (Liu et al., 2014)	Whitetail dascyllus	26 (Thresher, Colin & Bell, 1989)	Cytochrome-B (1058)	56	
			GluDG-L/H16460 (Palumbi et al., 1991)		
Lutjanus kasmira (Gaither et al., 2010)	Bluestripe snapper	60 (Baensch, 2014)	Cytochrome-B (446)	48	
			Cytb9/Cytb7 (DiBattista et al., 2013)		
Nerita plicata (Crandall et al., 2008)	Whorled nerite	~180 (Underwood, 1978)	Cytochrome Oxidase I (613)	50	
			LCO-1490/HCO-1498 (Folmer et al., 1994)		
Pomacentrus coelestis (Liu et al., 2012)	Neon damselfish	39 (Wilson & McCormick, 1999)	Control Region (337)	50	
			CR-A/CR-E (Lee et al., 1995)		

In the present study, we added sequence data from nine coral reef species sampled within the DANP to existing datasets in GEOME to test metapopulation hypotheses regarding the role of each species’ Dongsha population in the greater context of the Indo-Pacific. We then used a graph theoretical analysis based on maximum dispersal distances to more closely examine Dongha’s role as an intergenerational stepping-stone within the South China Sea.

Materials & methods

Sampling and sequencing

We selected nine target species from the Genomic Observatories Metadatabase (GEOME; Deck et al., 2017) based on the availability of genetic data from nearby populations and their common occurrence at Dongsha Atoll, including eight reef fishes and the intertidal gastropod Nerita plicata (Table 1). In March of 2017, tissue samples for these nine species were collected at Dongsha by SCUBA divers using spears in the case of reef fish species, or by hand in the case of the gastropod. Tissues were preserved in 95% ethanol. The field sampling of the present study is under the permit number 0000691 which was approved by the Marine National Park Headquarters in Taiwan.

Mitochondrial DNA (mtDNA) amplification and sequencing were conducted at California State University Monterey Bay as part of the Molecular Ecology and Evolution capstone research course in Fall of 2017. Marker choice was made based on the data available in GEOME and included the mitochondrial Cytochrome Oxidase I, Cytochrome-B, and the Control Region. In each case, published primers were used (see Table 1 for details). DNA was extracted in a 10% Chelex® (Biorad) solution following Walsh, Metzger & Higuchi (1991). Polymerase chain reactions (PCR) were conducted in 25 μL reactions with 2.5 μL of 10x PCR buffer, two μL MgCl2 (25 mM), 2.5 μL dNTPs (8 mM), 1.25 μL of each 10 mM primer, one μL of DNA template, and 0.625 U of AmpliTaq (Applied Biosystems, Waltham, MA, USA). Thermocycling conditions were the same across species, only differing in annealing temperature: initial denature for 2 min at 95 °C; followed by 35 cycles of denaturation at 95 °C for 15 s; annealing at T °C for 30 s (where T is given in Table 1); and elongation at 72 °C for 1 min; with a final elongation step at 72 °C for 7 min. PCR products were checked on a 1% agarose gel using GelRed (Biotium, Fremont, CA, USA). Successful PCR products were sent to the MCLab (South San Francisco; mclab.com) for cleanup, cycle sequencing (both directions), and sequencing on an ABI 3730 DNA Sequencer. Forward and reverse reads for each sample were proofread and aligned using Geneious 9.1.8 software. Complementary sequence data from four or five nearby populations were downloaded for each species from the GEOME database (Fig. 1). Sequences for each species were aligned and trimmed to a common length using the muscle algorithm with default parameters implemented in Geneious v9 (Biomatters) and exported to FASTA format.

Population genetic analysis

We used the pegas (Paradis, 2010) and strataG (Archer, Adams & Schneiders, 2017) population genetic packages to read the FASTA-formatted data into R and identify unique haplotypes. We then utilized these packages to estimate standard genetic diversity statistics for the Dongsha population of each species. Haplotype diversity and the number of haplotypes unique to Dongsha were calculated using the exptdHet, privateAlleles functions in strataG. Fu’s Fs, a statistic which identifies populations with an excess of recent substitution events caused by demographic growth, genetic hitchhiking, or background selection (Fu, 1997), was calculated using the fusFS functions in the same package. Nucleotide diversity was measured using the nuc.div function in pegas. The significance of Fs was determined with 1,000 coalescent simulations of neutrality in Arlequin 3.5 (Excoffier & Lischer, 2010). Haplotype diversity, nucleotide diversity, and percentage of private haplotypes (unique to a given population) were also averaged across all nearby sampled populations, and a two-sided t-test was used to determine if the Dongsha populations’ genetic diversity was significantly different from the nearest sampled populations.

Pairwise ΦST, a sequence analog of FST (Excoffier, Smouse & Quattro, 1992), was calculated with pairwiseTest in strataG with significance determined by 1,000 randomly drawn permutations of the data to represent the null hypothesis of no genetic structure. Finally, pairwise matrices were visualized in ggplot2 and as non-metric dimensional scaling (NMDS) plots to represent the distances in two-dimensional space using the metaMDSiter function in the R package vegan (Oksanen et al., 2017), using a hybrid model of monotone and linear regression for ΦST values lower than 0.1. For both visualizations, negative values were corrected to zero, and for NMDS, zero or negative values had a very small positive value added to them. To visualize whether there is a geographic pattern of haplotype distribution, we defined samples used in this study into six regional groups (Fig. 1). Median joining networks with these regional colors were created using PopArt (Leigh & Bryant, 2015). All analyses are detailed at https://github.com/ericcrandall/dongsha/.

We also estimated the marginal likelihood of three different metapopulation models for the nine species in Migrate-n 3.6.11 (Beerli & Felsenstein, 2001; Beerli & Palczewski, 2010; Fig. 2): (a) extremely high levels of larval dispersal (proportion of migrants (m) > 0.1) throughout the sampled range yielding effective panmixia, (b) slightly restricted larval dispersal (m < 0.1) represented by an n-island model with equal population sizes and equal migration between all population pairs, (c) restricted larval dispersal such that larvae are only exchanged between neighboring populations or regions as represented in a stepping-stone model (two populations were determined to be neighboring if no other sampled populations would serve as a likely intermediate stepping-stone).

Figure 2 Visualization of all metapopulation models tested for each species.

Visualization of all metapopulation models tested for each species. Black lines below each species name indicate a distance of 1,000 km. Blue text indicates sampled sites which are arranged in geographic space. Dotted lines with arrows connect every pair of sample sites and indicate directional migration (gene flow) parameters included in the n-island model while solid lines with arrows indicate directional migration parameters included in the stepping-stone model. The model of panmixia treated all sampled localities as a single population.

Migrate-n analysis followed methods developed in Crandall et al. (2019b). FASTA-formatted datasets were converted to Migrate-n format using PGDSpider 2.0.5.1 (Lischer & Excoffier, 2012). For each species, we found optimal parameters (gamma shape parameter, transition transversion ratio and base frequencies) for an HKY + G model of molecular evolution using the modelTest function in the R package phangorn (Schliep, 2011). The gamma shape parameter was discretized to four categories using the discrete.gamma function from the same package. All models had identical, windowed exponential priors on Θ (lower bound: 1 × 10−5, upper bound: 1 × 10−1, mean: 0.01) and m/μ (lower bound: 1 × 10−4, upper bound: 1 × 106, mean: 1 × 105) parameters. We used four heated chains with temperatures of 1, 1.5, 3, and 1 × 105 to ensure a thorough search of parameter space, thereby enabling an estimate of model marginal likelihood via path sampling (Beerli & Palczewski, 2010). Migrate-n was set to optimize on the m/μ parameter rather than the joint parameter Nem (to avoid correlations with the Θ (= Ne μ) parameters), and with an inheritance scalar that reflected the haploid, uniparental transmission of mtDNA. For each model, the coolest chain explored fifty million genealogies, sampling every 500 iterations, and discarding the first five million genealogies as burn-in.

Each Migrate-n run comprised three replicates of fifty million genealogies to yield a single estimate of marginal likelihood. Each run was then repeated three times, to yield three independent estimates of metapopulation model marginal likelihood from nine replicate runs. The Bezier-corrected estimate of model marginal likelihood (which approximates the marginal likelihood from a larger number of heated chains) was harvested from each outfile and averaged across the three independent replicate runs. Mean Bezier-corrected marginal likelihoods were converted to relative model probabilities and Bayes factors following Johnson & Omland (2004). To account for variance in mean marginal likelihoods across the three replicate runs, a permutation t-test was run to compare the mean marginal likelihoods of the first and second ranked models, following Crandall et al. (2019b). Parameter files for each species and each model, as well as code for interpreting the output, are available in the Github repository, within the migrate_analysis directory.

We then asked to what extent the models selected for each species by Migrate-n were a product of the maximum pelagic larval duration (PLD), especially given the spatially heterogeneous sampling of surrounding populations. Because only two of three possible models were selected by all nine species, we constructed a logistic regression model using values for maximum PLDs from the literature (Table 1) and great-circle distances from Dongsha Atoll to both the nearest and furthest sampled populations for each species, calculated with the Raster package (Hijmans, 2021) in R: (p(Migrate Model) ~ Maximum PLD + Closest Distance + Furthest Distance). We used backward BIC model selection in the R package MASS (Venables & Ripley, 2002) to select the best model.

Graph theoretical analysis

To better place our results in the geographical context of the South China Sea, we undertook a graph theoretical analysis of the potential for larval dispersal in this region. We developed two rasters (Fig. 1B) representing (1) coral reef areas and (2) land areas of continents and islands of the South China Sea from Fig. 1 of Zhao et al. (2016) and Fig. 1A of Dorman et al. (2016), and projected both rasters into UTM coordinates (zone 50N) with 1 km resolution using the raster package. We then imported these rasters into Graphab 2.6 (Foltête, Clauzel & Vuidel, 2012), defining the reef areas with at least 25 hectares as habitat patches, and the land areas as a cost raster with each pixel costing 10,000. Graphab then generated a linkset between reef patches for each species using a maximum dispersal distance calculated as the product of maximum PLD (Table 1) and the mean current speed of 18.7 km/day given by Hu et al. (2000). The basic binary simple (undirected) graph generated by Graphab from this linkset was then imported into the igraph package for R (Csardi & Nepusz, 2006), which we used to measure betweenness centrality, defined as the fraction of shortest paths between all patches which pass through a given reef patch. Estrada & Bodin (2008) have shown that betweenness centrality is an important measure of the overall importance of a patch to the large-scale connectivity of a landscape. We also used igraph to measure the diameter of each species’ graph, as the minimum number of larval-dispersal steps connecting the two most distant points. It is important to emphasize that this approach is not equivalent to a biophysical model of larval dispersal such as found in Dorman et al. (2016), and most dispersal is expected to occur at much shorter distances (Cowen et al., 2000). However, since genetic structure is very sensitive to dispersal at the tails of the larval dispersal kernel (Grosberg & Cunningham, 2001), this approach might provide an approximate understanding of potential gene flow in the South China Sea.

Results

We sequenced a total of 206 tissue samples from Dongsha across the nine species. Sequence length varied from 316 bp (Control Region, Ctenochaetus striatus) to 1,058 bp (Cytochrome-B, Dascyllus aruanus). A total of 111 unique haplotypes across all nine species were detected (Table 2). Within our dataset, 58% of these haplotypes were apparently private to Dongsha Atoll (at least within the local region sampled from GEOME), ranging from 5% in the oval butterflyfish Chaetodon lunulatus to ~80% in the whorled nerite Nerita plicata and the striated surgeonfish Ctenochaetus striatus. The bluestripe snapper Lutjanus kasmira had significantly elevated genetic diversity (h, π and % private alleles) at Dongsha compared to the rest of the region; however, this may be an artifact of low sample size. The neon damselfish Pomacentrus coelestis had significantly lower genetic diversity (h, π, and % private alleles) at Dongsha than the surrounding populations. All but three species had Fu’s FS values that were significantly negative at the recommended alpha of p < 0.02 (Fu, 1997), indicating a departure from neutrality due to an excess of recent mutations at the tips of the genealogy. Median-joining haplotype networks showed no clear geographic pattern of haplotype distribution for any species (Fig. 3). In addition, all haplotype networks showed a star-like topology providing visual confirmation of the low Fu’s FS values and evidence of either recent natural selection or population expansion. The combined data set used in this study has been shared with location and date metadata in GEOME (geome-db.org) within the “Reef Species of Dongsha Atoll” expedition of the Diversity of the Indo-Pacific Project, with GUID: https://n2t.net/ark:/21547/Dos2. By clicking “Query All Reef Species of Dongsha Atoll Samples”, the fasta file can be downloaded through the download option next to “map” option.

Figure 3 Haplotype networks of nine species.

Median-joining networks for all nine species. Each circle represents a haplotype, with the frequency of the haplotype indicated by the circle’s size (scale varies across species). Pie charts indicate each haplotype’s distribution across sampling sites. Lines indicate possible mutational changes between haplotypes, with hash marks representing more than one change.

Table 2 Genetic diversity statistics including haplotype diversity (h), nucleotide diversity (π) and Fu’s FS for each sampled Dongsha population in comparison to regional means.

Species	Dongsha N	# Haplotypes	Private haplotypes	% Private	Regional mean % private	h	Regional mean h	π	Regional mean π	Fs	
A. japonicus	14	9	3	21.43	12.96	0.88	0.86	0.006	0.004	−3.53	
C. auriga	28	6	2	7.14	7.06	0.51	0.56	0.001	0.001	−4.16	
C. lunulatus	19	4	1	5.26	13.92	0.56	0.66	0.004	0.005	2.19	
C. striatus	26	24	21	80.77	75.03	0.99	0.99	0.025	0.023	−15.44	
C. vrolikii	24	12	9	37.50	29.37	0.79	0.80	0.003	0.003	−7.45	
D. aruanus	44	13	5	11.36	14.63	0.79	0.75	0.002	0.002	−5.19	
L. kasmira	6	6	4	66.67	16.20**	1*	0.59	0.007**	0.002	−3.03	
N. plicata	24	22	19	79.17	77.98	0.99	1.00	0.011	0.013	−14.99	
P. coelestis	21	15	9	42.86	63.94**	0.90**	1.00	0.010**	0.014	−10.07	
Note:

Significant deviations from regional means are noted at *p < 0.05 (*) and **p < 0.01. Significantly low FS values (compared to neutral coalescent simulations) at p < 0.02 are denoted in bold.

Pairwise ΦST values varied from 0 (found in every dataset) to 0.202 in different taxa across Indo-Pacific (Fig. S1). In the Japanese surgeonfish, Acanthurus japonicas significant genetic structures were found between Okinoerabu (near Okinawa Island), Philippines, and Xisha. In addition, the Dongsha population of L. kasmira was highly and significantly structured with all other populations. However, low sample size in the Okinoerabu population of A. japonicus (n = 6) and the Dongsha population of L. kasmira (n = 6) curtails our ability to interpret the significance of these observations of high ΦST. The only other significant value of ΦST (0.077) for a Dongsha population was with the Xisha Islands in the whitetail dascyllus, Dascyllus aruanus. On average, the Dongsha population of each species had 1.9 positive pairwise ΦST values out of a total of 3 or 4 populations that it was measured against. Due to the lack of co-sampling of each species at these other sites, there were no clear geographic patterns in Dongsha’s genetic structuring with other populations. NMDS plots (Fig. S2) generally showed a similar lack of correlation with geography, except for perhaps C. striatus and C. lunulatus. This general lack of geographic signal in genetic structure is a common feature in the Indo-Pacific (Gaither et al., 2011; Crandall et al., 2019a) and marine population genetic datasets in general (Selkoe & Toonen, 2011; Selkoe et al., 2016).

Relative probabilities for each of three metapopulation models tested by Migrate-n are depicted in Fig. 4. Following guidelines laid out by Kass & Raftery (1995), Migrate-n found strong support for a metapopulation model wherein the Dongsha population acts as an important regional stepping-stone for oval butterflyfish (Chaetodon lunulatus; 5.42 × 109: 1 odds against panmixia; see Table 3), striated surgeonfish (Ctenochaetus striatus; 1.78 × 108: 1 odds against panmixia; see Table 3, Fig. 4), and substantial support for a stepping-stone model in the pearlscale angelfish (Centropyge vroliki; 12:1 odds against panmixia). The whitetail damselfish (Dascyllus aruanus) had modest support for a stepping-stone model (4:1 odds against panmixia), but the mean log-likelihood for this model was not significantly higher than that for panmixia (p = 0.15), leaving this result ambiguous. Datasets from the five other species strongly and unambiguously supported a regional model of effective panmixia (Table 3).

Figure 4 Relative probability of each of four metapopulation hypotheses.

Relative probability of each of four metapopulation hypotheses depicted in Fig. 2 for each of nine species sampled at Dongsha as calculated from Migrate-n marginal likelihoods averaged across three replicate runs.

Table 3 Most probable (1°) and second most probable (2°) models and their relative probabilities for each species, followed by the P-value of a one-tailed permutation t-test of the alternate hypothesis that the mean ln-likelihood of the 1° model is significantly higher than the mean of the 2° model.

Species	1° Model	1° Probability	2° Model	2° Probability	p 1° Mean > 2° Mean	2 Ln Bayes Factor 1°/2°	Odds 1°:2°	
A. japonicus	Panmixia	0.992	Stepping-stone	0.008	0.05	9.55*	118.4:1	
C. auriga	Panmixia	1.000	Stepping-stone	0.000	0.05	19.29*	1.54 × 104:1	
C. lunulatus	Stepping-stone	1.000	Panmixia	0.000	0.05	44.83*	5.42 × 109:1	
C. striatus	Stepping-stone	1.000	Panmixia	0.000	0.05	37.99*	1.78 × 108:1	
C. vrolikii	Stepping-stone	0.921	Panmixia	0.079	0.05	4.90	11.6:1	
D. aruanus	Stepping-stone	0.742	Panmixia	0.169	0.15	2.96	4.4:1	
L. kasmira	Panmixia	1.000	Stepping-stone	0.000	0.05	441.07*	6.0 × 1095:1	
N. plicata	Panmixia	1.000	Stepping-stone	0.000	0.05	15.92*	2.86 × 103:1	
P. coelestis	Panmixia	1.000	Stepping-stone	0.000	0.05	123.10*	5.37 × 1026:1	
Note:

Loge Bayes Factor indicates the relative probability of the best model relative to the second best model, with values greater than six indicating a strong weight of evidence (odds > 20:1, indicated with *), and values greater than three indicating substantial support (bolded, Kass & Raftery, 1995). P-values indicate the outcome of a permutation t-test comparing the log-likelihoods of the two top-ranked models across three replicated Migrate-n runs, bolded at alpha of 0.05.

Backward stepwise model selection found that maximal and minimal distances between Dongsha and other sampled populations were not important predictors of whether Migrate-n selected a stepping-stone model. Maximum PLD was the only important predictor in explaining when Migrate-n selected a stepping-stone model over panmixia, with an improvement of 4.12 units of log-likelihood between the full model and one with PLD only. The resultant model was significant (p = 0.0473), with the coefficient indicating that for every day increase in PLD, there is a 9.3% decrease in the odds of selecting a stepping-stone model (95% CI [0.002–23]; Fig. 5).

Figure 5 Logistic regression model.

Logistic regression model for: Migrate-n model ~ maximum PLD, with 95% confidence intervals constructed as 1.96 × standard error. The model is shown between 20 and 70 days larval duration to avoid extrapolation. Black circles show species with datasets that selected a stepping-stone model, while open circles show species datasets that selected a model of effective panmixia.

Graphab found 115 reef patches greater than 25 hectares in the South China Sea. These patches were connected with a minimum graph diameter of one larval dispersal step (Nerita plicata) or a maximum of seven generational steps for Dascyllus aruanus (Fig. 6). Graph diameter was highly correlated with PLD (r2 = 0.61), so we do not present a separate logistic regression for its ability to predict whether a species had a stepping-stone model. Out of 115 reef patches, Dongsha was always ranked in the top 15 reef patches for betweenness centrality, ranging from the 3rd most important reef (Chaetodon lunulatus and Pomacentrus coelestis) and the 14th most important reef (Dascyllus aruanus) for this metric. The only reefs with consistently higher rankings were found in the Xisha Islands, the Zhongsha Islands, and Huangyan Island (Fig. 6).

Figure 6 Undirected binary simple graphs between 115 South China Sea reef patches with areas greater than 25 hectares.

Circle sizes are proportional to the approximate reef area of each patch, while darker colors indicate a higher relative betweenness centrality for a given patch. Blue triangles indicate a genetic sample for that species. Yellow lines show shortest overwater paths between reef patches that could potentially be connected by larval dispersal assuming a maximum distance given in the lower right corner. Additional statistics in the lower right corner include pelagic larval duration (PLD), graph diameter (D) as the shortest number of larval dispersal event required to cross the longest distance between reef patches for a given species and the ranking of betweenness centrality (BC) for Dongsha out of 115 reef patches. Genetic data for species with underlined names supported a stepping-stone model of dispersal.

Discussion

While this genetic analysis of nine coral reef species represents a relatively small proportion of the thousands of species comprising the Dongsha Atoll National Park community, it is also the most extensive analysis attempted to put a protected single reef community into the broader ecological and evolutionary context of the Indo-Pacific to date. Overall, our analysis indicates that over evolutionary timescales, Dongsha populations are well-connected with the rest of the South China Sea and indeed with the rest of the Indo-Pacific and maintain genetic diversities that do not significantly deviate from average (except for P. coelestis, see discussion below). Furthermore, for marine species with mean PLDs less than 40 days (the vast majority; Strathman, 1987; Shanks, 2009), Dongsha Atoll provides a valuable stepping-stone that is consistently within the top 15 reef patches for promoting regional genetic and demographic connectivity within the South China Sea (Figs. 5 and 6).

Genetic population structure between Dongsha Atoll and nearby populations, as measured by the sequence-based FST analog ΦST, was generally low and non-significant (Fig. S1). This finding is congruent with the results of connectivity studies targeting damselfishes between the SCS and Kuroshio regions which showed low and non-significant genetic structure among the Xisha (Paracel) Islands, Dongsha, and Taiwan (Liu et al., 2011, 2014, 2019). Failure to reject panmixia is generally common in marine species (Waples, 1998; Kinlan & Gaines, 2003), has been found consistently in the Indo-Pacific (Keyse et al., 2014; Crandall et al., 2019a), and is traditionally interpreted as evidence for relatively high levels of gene flow due to larval dispersal. However, these summary statistics can often be zero even when there has not been gene flow in thousands of generations (Faurby & Barber, 2012; Crandall et al., 2019a), and they do not provide any information about metapopulation structure.

Five of the nine species surveyed here demonstrate strong regional connectivity in the South China Sea such that their gene pools are well-mixed (effective panmixia; mean SCS graph diameter for these species = 3.0 intergenerational dispersal events, sd = 1.22). In contrast, the coalescent approach employed here clearly distinguishes these from the four species that have some regional metapopulation structure (stepping-stone; Table 3, Fig. 4). This latter group generally has shorter PLDs (mean PLD = 37 days, sd = 15 days) than the former group (mean PLD = 78 days, sd = 60 days), meaning that it will take more generations to cross the SCS (mean SCS graph diameter for these species = 5.25 intergenerational dispersal events, sd = 1.70). For these shorter PLD species, Dongsha Atoll provides an important intergenerational stepping-stone for maintaining gene flow across the SCS (highly ranked betweenness centrality for all species except N. plicata; Fig. 6). Furthermore, Fig. 5 demonstrates a significant relationship between PLD and metapopulation structure (albeit with wide confidence intervals) with a sharply increasing probability that a species will have a metapopulation structure that relies on Dongsha Atoll as PLD decreases to about 40 days or less. Below, we discuss the oceanographic context of the SCS as it relates to our findings.

Oceanographic circulation patterns in the SCS and larval dispersal

The seasonal circulation patterns in the SCS are mainly driven by monsoon winds and comprise several cyclonic/anticyclonic eddies. In contrast, northeastward monsoons and southwestward monsoons prevail during winter and summer, respectively (Hu et al., 2000, Fig. 1B). During winter, most currents are northeastward and turn east of Natuna Islands toward the west coast of Luzon in the Philippines. Meanwhile, the Kuroshio intrusion splits into two currents, with one branch moving toward Dongsha and another toward Xisha Islands (Paracel Islands), then turns northward to pass along the Taiwan coast. During summer, most of the currents flow southwestward, with a mean current velocity of 18.7 km/day, while the southwestward monsoon weakens the Kuroshio Intrusion.

Given this circulation pattern, Fig. 6 provides heuristic estimates of the maximum dispersal distances for larvae of each species during the summer, when most corals and fish are spawning (Liu, 2011; Ho, 2017). For example, even larvae from the species with the shortest PLD (Dascyllus aruanus; 26 days) should be able to drift to the nearest reefs 230 km away near Hong Kong, or to Taiwan, 433 km away (D. aruanus; Figs. 1B and 6). In contrast, species with the longest PLD (Nerita plicata; 180 days) should be able to reach anywhere in the SCS in a single larval dispersal event. These estimates are heuristic because the vast majority of larvae released at Dongsha or elsewhere will not be advected over the maximal distance due to diffusion and mortality (Cowen et al., 2000). On the other hand, it is possible that larvae may occasionally be advected beyond the depicted maximum distance by infrequent events such as typhoons or by the extension of their larval duration beyond what has been measured (McCormick, 1999). In the present study, Zongsha seems to act as an important hub for the connectivity in the area between Xisha, Zhongsha and Huangyang Island as revealed in Fig. 6. Liu, Hsin & Cheng (2020) suggested that the seasonal dynamic of eddies may facilitate the dispersal of marine organisms in this area which may support the role of Zongsha in term of population connectivity. Meanwhile, the heuristic estimates in Figs. 1B and 6 are useful in showing that, in general, the species for which a stepping-stone model of metapopulation structure was selected are only able to reach neighboring reefs, while those for which panmixia was selected have the capacity to disperse widely in the SCS and beyond.

Moreover, more sophisticated biophysical modeling of larval dispersal in the South China Sea indicates a clear regional structure for species with a PLD of 40 days or less (Melbourne-Thomas et al., 2011; Dorman et al., 2016; Liu, Hsin & Cheng, 2020). Although focused on the central and southern SCS, Melbourne-Thomas et al. (2011) modeled particles with 30 or 40 day PLD, active or passive dispersal with a rate of mortality of 0.1 to 0.2 per day, and showed that all particles released from the Nansha Islands (i.e., Spratly or Kalayaan islands) settled either in the Nanshas or in western Palawan. Using more particles, slightly more realistic biological parameters, and tracking particles released from the Nanshas for 90 days or until settlement, Dorman et al. (2016) got similar results, with most particles staying in the Nanshas or moving eastward to Palawan, but they also showed limited connectivity from the Nanshas to the Xishas and Northern Luzon, especially during the Fall when currents shift to the northeast. Finally, Liu, Hsin & Cheng (2020) simulated benthic currents at 200-400 m depth over 60 days to show little connectivity between deep-sea coral (Deltocyathus magnificus) populations near Dongsha and the Xisha Islands, which was confirmed by clear genetic structure in microsatellites.

The significant relationship between PLD and the existence of metapopulation structure demonstrated here (Fig. 5) has wide confidence intervals due to two species. A metapopulation model of effective panmixia was selected for the neon damselfish, Pomacentrus coelestis despite having a maximum PLD of 39 days, while a stepping-stone model was most probable for C. striatus, even though it has a maximum PLD of 59 days (Wilson & McCormick, 1999). Although the best model for explaining metapopulation structure did not include sampling distance as a factor, we suggest that heterogeneous sampling may have played a role in the models that were selected for at least these two species. The nearest sampled population to Dongsha for the P. coelestis dataset was 411 km away in Taiwan, while the furthest was 1,304 km away in Okinawa. Furthermore, the fact that the Dongsha sample of P. coelestis deviated from average regional diversities suggests that our sample of this species may be non-representative. The low genetic diversities in Table 2 may be due to cohesive dispersal among related larvae (Robitzch, Saenz-Agudelo & Berumen, 2020) since the specimens that we collected were juveniles around the same size and collected at the same location. Therefore, the haplotypes identified from these samples may derive from a relatively small number of parents. For C. striatus, even the nearest sampled population was much further than Okinawa, 2,280 km away in Bunaken near Sulawesi, while the furthest sampled population was 3,217 km away near Krakatau in the Sunda Strait. Given these heterogeneous sampling distances and the possibly non-representative sample of P. coelestis, it is easier to understand how a stepping-stone model was inferred over large distances for C. striatus while a model of panmixia was selected over short distances for P. coelestis. Future efforts to comparatively model metapopulation structure should standardize sampling to the extent that it is possible.

Conclusions

For the relatively low cost of adding mitochondrial sequence data from nine coral reef species sampled within the national park at Dongsha Atoll to existing datasets we were able to successfully test metapopulation hypotheses of larval dispersal and gene flow for each of these reef species. While it is important to acknowledge that these results derive from only a single genetic locus, results from initial mitochondrial surveys are often borne out by multi-locus analyses (Bowen et al., 2014). Our results from both population genetic and graph theoretical analysis demonstrate that Dongsha is likely a key stepping-stone for promoting genetic and demographic connectivity among reefs in the northern South China Sea, especially for species with PLDs less than 40 days. Therefore, reinforcement of the management by Marine National Park Headquarters is crucial to reduce the fishing pressure (i.e., illegal poaching) and maintain Dongsha populations. Meanwhile, research efforts need to be increased to better understand the role of Dongsha Atoll in connectivity partners across the region. Missing from these analyses are the habitat-building scleractinian corals and seagrasses that are essential to the long-term health of these threatened ecosystems. A comprehensive program to sample and sequence coral reef species throughout the South China Sea could provide a more detailed and empirical understanding of the valuable protections afforded by protected areas such as Dongsha Atoll.

Supplemental Information

Supplemental Information 1 ΦST values for pairwise comparisons between all sampled populations for each of nine species.

ΦST values for pairwise comparisons between all sampled populations for each of nine species. * indicates values that are significant at p < 0.05, and ** indicates values that are significant at p < 0.01. Darker shades of red indicate stronger genetic structure

Click here for additional data file.

Supplemental Information 2 Non-metric dimensional scaling (NMDS) plots of nine species.

Non-metric dimensional scaling (NMDS) results based on FST matrices in Fig. S2 depict Dongsha’s genetic relationship with neighboring populations in two-dimensional space for all nine species.

Click here for additional data file.

We thank the technicians of the Dongsha Atoll Research Station (DARS), as well as Margaret Geissler and Melanie Chu at CSU Monterey Bay for their assistance in the field and laboratory and for logistical support. We thank Paul Barber for sharing previously unpublished data for C. striatus into GEOME.

Additional Information and Declarations

Competing Interests

Author Contributions

Animal Ethics

Field Study Permissions

Data Availability

Robert J. Toonen is an Academic Editor for PeerJ.

Shang Yin Vanson Liu conceived and designed the experiments, performed the experiments, analyzed the data, prepared figures and/or tables, authored or reviewed drafts of the paper, and approved the final draft.

Jacob Green performed the experiments, analyzed the data, prepared figures and/or tables, authored or reviewed drafts of the paper, and approved the final draft.

Dana Briggs performed the experiments, analyzed the data, prepared figures and/or tables, authored or reviewed drafts of the paper, and approved the final draft.

Ruth Hastings performed the experiments, analyzed the data, prepared figures and/or tables, authored or reviewed drafts of the paper, and approved the final draft.

Ylva Jondelius performed the experiments, analyzed the data, prepared figures and/or tables, authored or reviewed drafts of the paper, and approved the final draft.

Skylar Kensinger performed the experiments, analyzed the data, prepared figures and/or tables, authored or reviewed drafts of the paper, and approved the final draft.

Hannah Leever performed the experiments, analyzed the data, prepared figures and/or tables, authored or reviewed drafts of the paper, and approved the final draft.

Sophia Santos performed the experiments, analyzed the data, prepared figures and/or tables, authored or reviewed drafts of the paper, and approved the final draft.

Trevor Throne performed the experiments, analyzed the data, prepared figures and/or tables, authored or reviewed drafts of the paper, and approved the final draft.

Chi Cheng performed the experiments, analyzed the data, prepared figures and/or tables, authored or reviewed drafts of the paper, and approved the final draft.

Hawis Madduppa performed the experiments, authored or reviewed drafts of the paper, and approved the final draft.

Robert J. Toonen conceived and designed the experiments, performed the experiments, authored or reviewed drafts of the paper, and approved the final draft.

Michelle R. Gaither conceived and designed the experiments, performed the experiments, authored or reviewed drafts of the paper, and approved the final draft.

Eric D. Crandall conceived and designed the experiments, performed the experiments, analyzed the data, prepared figures and/or tables, authored or reviewed drafts of the paper, and approved the final draft.

The following information was supplied relating to ethical approvals (i.e., approving body and any reference numbers):

The species used in this study are non-regulated animals. Therefore, there is no relevant provision for the ethical review process in Taiwan.

The following information was supplied relating to field study approvals (i.e., approving body and any reference numbers):

Field experiments were approved by the Marine National Park Headquarters (Taiwan) under permit number 0000691.

The following information was supplied regarding data availability:

The combined data set used in this study has been shared with location and date metadata in GEOME (geome-db.org) within the “Reef Species of Dongsha Atoll” expedition of the Diversity of the Indo-Pacific Project, with GUID, available at: https://n2t.net/ark:/21547/Dos2. By clicking “Query All Reef Species of Dongsha Atoll Samples”, the FASTA file can be downloaded through the download option next to “map” option.

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
