# Peer review of "Dongsha Atoll is an important stepping-stone that promotes regional genetic connectivity in the South China Sea"

_PeerJ, doi:10.7717/peerj.12063_

## Round 0.1 · original submission · Major Revisions

Two peer reviewers provided excellent and thoughtful reviews of this paper and they, as well as myself, applaud the broad taxonomic range of species included in this analysis.

Both reviewers and I have some concerns about the overall conclusions of this study, which note that Dongsha Atoll is an important stepping-stone, though it is not clear between which locations it is important as a stepping stone. The figures and analyses appear to indicate that several locations are important stepping-stones, with no real emphasis on Dongsha Atoll. Reviewer 2 suggests removing Dongsha Atoll from the analyses to test how the model results change with and without this location. Finally, there is mention in some locations of five Migrate models when only three were tested in the actual paper.

Reviewer 1 notes that some species show a high proportion of private haplotypes at Dongsha, which could note its importance for marine conservation.

Reviewer 2 also notes that they were unable to access the raw sequence files to assess the data, and Supplemental files could not be opened. Please look into this so that all data and files are accessible.

Reviewer 1 ·

Basic reporting

Dongsha Atoll is an important stepping-stone that promotes regional genetic connectivity in the South China Sea described by Liu et al. examined meta-population model of 9 marine species using mitoDNA and coalescent based simulation and found for species with relatively short larval duration (<40 days), stepping stone model is best supported and thus concluded Dongsha Atoll is an important place for regional connectivity. By applying coalescent based analysis, authors overcome the small number of samples per species and population that exist for traditional population based analysis. The manuscript is overall very well written and analyzed rigorously utilizing GEOME data set. The results obtained here are very important and useful for conservation of Dongsha Atoll and its surrounding region.
Adding some more information, I think the paper would be more understandable for non-specialist.

1) It was not clear for me, Dongsha Atoll is in particularly important stepping-stone for which 'regions' in the SCS (from where to where)?

2) What is the relative importance of Dongsha Atoll for sustaining SCS meta-populations? It also seems all the populations are important as they are connected by stepping-stone.

3) Fig2 and L121-L128 it seems only 3 models are tested not 5. Please explain in more detail. At least, it was not possible to identify 5 different models by looking at Fig.2.

4) Because this study focus only on mtDNA data set, please add some small notes that none of the target species suffer from ancient introgressive hybridization or sex biased dispersal etc.

5) for some species such as whorled nerite and the striated surgeonfish who had high proportion of private haplotypes, Dongsha Atoll is of importance for conservation of genetically unique population (evolutionary novelty) rather than just a stepping stone for conservation. It means the Atoll is isolated for relatively long time (evolutionary time scale rather than ecological time scale in terms of conservation).

Experimental design

Research question testing the hypothesis the Dongsha act as a stepping stone for regional connectivity is clearly stated.
Rigorous investigation performed to a high technical & ethical standard.

L106 please clarify what do you mean by " the surrounding area"
L140 please briefly describe why optimized on the m/u instead of Nem

Validity of the findings

Meaningful replications (3 independent runs for each coalescent analysis) using 9 species with different PLD were assessed.
The result and conclusion is robust and scientifically sound.
Conclusion are well stated but please more clarify the three points I wrote in the basic reports.
Limitation is it only based on mtDNA not multilocus genomic data.

Additional comments

The paper is of high scientific and conservation value.
Keep up a good work utilizing GEMOE data base.

Reviewer 2 ·

Basic reporting

This manuscript is well-written and generally clear with a few small typos (see below). The final two paragraphs are duplicates and one needs to be deleted.
There are two major issues with the data presented:
1. I could not access the raw data/alignment files following the directions provided (and some probing). So I could not assess this part of the paper.
2. One of the supplemental files (Figure S2) I could not open despite trying multiple programs. Converting to a more common file type would be recommended. Again I could not assess this major part of the results.

Line 3 – semi colon between references
Line 17 - semi colon between references
Line 299- semi colon between references

Line 195 – This is the first time Okinoerabu is used in the text, and it does not match any of the locations in the map in Fig.1. I think it is somewhat intuitive that it is Okinawa, but consistency with the map or an explanation that it represents Okinawa would be helpful.

Figure S2 - The combined axes for all 9 species made this confusing to read and interpret, as the axes are not all equivalent. Separating them out would add clarity for the reader.

Experimental design

The research question is well-defined and clearly filling a knowledge gap, and I applaud the authors for taking such a broad taxonomic and geographic overview of this area.
The methods are generally sufficiently detailed and there is enough information to replicate, but there are a couple of issues.

Most importantly: Paragraph starting at Line 121 – It appears that the authors have reduced the number of population models they were including in their paper. The text says 5 but only lists 3, and later only shows 3. And the github repository link shows results for both 3 and 5 different model scenarios. Please correct the text to the 3 scenarios described. I think it would clarify things for the reader to use the same Panmixia/N-island/Stepping stone phrasing more clearly here in the methods, so that it matches the results.

Lines 89-91 – How was the alignment carried out, was it using the geneious alignment software or clustal/muscle plug in, and under what parameters? Please elaborate.

Validity of the findings

My main concern with this paper is whether or not their approach actually answers their question.

The science they did appears sound and interesting. However, the main premise of the paper centers on the significance of Dongsha atoll with respect to regional connectivity, but their migrate-n models do not place any special significance on Dongsha (as far as I can see). It appears from the linked github page that the authors originally ran some other scenarios which specifically highlighted the role of Donghsa atoll, but they are not included in this paper. Data that support the main finding that Dongsha atoll is key for connectivity for the stepping-stone model species are not shown. It appears from the diagrams in Fig 4 that Dongsha is not a key part of the stepping stone model for all of the “stepping stone” species, for example: Chaetodon auriga seems to show Palau as the key stepping stone location, and for Chaetodon striatus Dongsha is somewhat peripheral.

The authors could try some extra analyses removing Dongsha atoll as a location from the models and seeing if connectivity significantly declines or something to that effect. But at the moment the authors do not present clear evidence for the statement “Dongsha Atoll provides an important intergenerational stepping-stone for maintaining gene flow across the SCS” (line 263). They show evidence for a stepping stone model of connectivity but their current approach cannot distinguish the importance of Dongsha relative to the other sampling locations.


Line 262 – The means and standard deviations of PLD between the two sets of very wide, and overlap substantially. Are the values stastically different? Please state significance in the text.

---

## Round 0.2 · Minor Revisions

One reviewer was able to provide a second review of this manuscript and all major comments have been addressed by the authors. Some minor edits remain which I agree with that should be addressed before acceptance including updating figure captions and making sure all supplementary files are accessible, though I also commend the authors for their comprehensive revisions for this paper.

Reviewer 2 ·

Basic reporting

All previous comments have been addressed. With the exception of the file for S2. I don't have a mac or adobe illustrator, so opening an .esp file is challenging. I still believe a more common file type to me more appropriate such as .pdf, or .svg, or something that doesn't need proprietary software or a file converter. I still have not been able to assess this part of the results.

I think Figure 6 could benefit from having Dongsha atoll highlighted in some way, so the reader doesn't have to refer back to previous maps. I would also suggest adding a total number of reefs analyzed to the figure (e.g. Dongsha rank = 10/112), so the reader understands the relative importance of 10th rank.

Experimental design

The authors gave the appropriate alignment program and settings in their response to authors but have not actually added it in the revised text, please do so.

The caption for Figure 4 still refers to 5 models. Please fix this.

All other comments were addressed well.

Validity of the findings

I commend the authors for their new analysis which does answer their research question about the importance of Dongsha atoll specifically. I think this highlights that Dongsha is one of the key connecting reefs in the area. It might be worth commenting on Zhongsha reef as this seems the most strikingly important reef from Figure 6. The authors comment on this in the results, but personal preference as to whether it deserves more discussion.

---

## Round 0.3 · accepted · Accept

The authors have done an excellent job of addressing reviewers' comments and suggestions to improve the manuscript, and I now believe this paper can be accepted.